# Establishment of an *in vivo* analytical method for detecting total anti-UFH activity and pharmacokinetic study in PS and R15 in rats

Huimin Li[1,2☯], Tong Li[3☯], Zhiyun Meng[2], Xiaoxia Zhu[2], Ruolan Gu[2], Hui Gan[2], Guifang Dou[2]*

1 Army 953 Hospital, Shigatse Branch of Xinqiao Hospital, Army Medical University (Third Military Medical University), Shigatse, China, 2 Department of Pharmaceutical Sciences, Beijing Institute of Radiation Medicine, Beijing, China, 3 State Key Laboratory of Bioactive Substance and Function of Natural Medicines, Institute of Materia Medica, Chinese Academy of Medical Sciences and Peking Union Medical College, Beijing, China

☯ These authors contributed equally to this work.
* tiaoji2013@163.com

## Abstract

Protamine sulfate (PS), the only U.S. Food and Drug Administration (FDA)-approved heparin antagonist, is encumbered by several drawbacks. R15, a synthetic polyarginine peptide, has proven to be a promising protamine substitute in prior studies. PS and R15 undergo biotransformation to active metabolites, underscoring the need for an analytical method that quantifies their total anti-heparin activity *in vivo*. Here, we reported the development and validation of such a method and described the pharmacokinetic profiles of PS and R15 in rats. Total anti-heparin activity in plasma was quantified by fortifying each sample with a fixed concentration of heparin and subsequently measuring the residual heparin. The method was fully validated for PS and R15 in accordance with Chinese bioanalytical guidance from Chinese Pharmacopoeia, confirming acceptable selectivity, precision and accuracy, stability, and dilution integrity. Pharmacokinetic profiles were then characterized in rats following single intravenous bolus administrations of PS at 300 U·kg$^{-1}$ and R15 at 300, 900, and 2700 U·kg$^{-1}$. An assay for quantifying total anti-heparin activity in rat plasma was successfully validated for both PS and R15. After a single intravenous dose of 300 U·kg$^{-1}$, R15 sustained anti-heparin activity for a markedly longer period (51.93 min *vs.* 3.94 min) and achieved an 18-fold higher of areaunder the curves (AUC = 632 min·µg·mL$^{-1}$ *vs.* 35.89 min·µg·mL$^{-1}$) with 19-fold higher mean residence time (MRT = 54.95 min *vs.* 2.59 min). Clearance (CL) for R15 and PS was 2.73 mL·min$^{-1}$·kg$^{-1}$ *vs.* 53.65 mL·min$^{-1}$·kg$^{-1}$, whereas the apparent volume of distribution ($V_d$) was of similar level (194 mL·kg$^{-1}$ *vs.* 268 mL·kg$^{-1}$), consistent with limited tissue distribution and prolonged intravascular retention. The extended exposure afforded by R15 is clinically advantageous because it mitigates the well-documented "heparin rebound"

**Data availability statement:** All relevant data are within the manuscript and its Supporting information files.

**Funding:** The author(s) received no specific funding for this work.

**Competing interests:** The authors have declared that no competing interests exist.

observed after rapid protamine clearance, thereby reducing the need for repeat dosing. R15 exhibited dose-dependent nonlinear pharmacokinetics, demonstrating saturable elimination processes typical of nonlinear pharmacokinetics. The validated assay, coupled with the *in vivo* rat pharmacokinetic study, provides a solid foundation for advancing R15's preclinical development.

## Introduction

Unfractionated heparin (UFH) is an anionic polysaccharide with an average molecular weight of 15–19 kDa [1,2], primarily used for systemic anticoagulation during surgery and for the prevention of thrombosis [1,3,4]. However, the use of UFH is often associated with risks of bleeding and other adverse reactions. Despite being largely replaced by low molecular weight heparins, recent reports indicate that the European heparin market is expected to be around $2 billion, potentially reaching $3 billion by 2022, with UFH accounting for approximately 10% of this market [5].

PS is a cationic protein mixture with a molecular weight of 4000–6000 Da, and is the only heparin antagonist approved for clinical use [6]. PS is primarily extracted from the sperm of wild salmon species, and this cationic protein binds to the anionic heparin, forming a UFH-PS complex that neutralizes the activity of heparin [7–10]. However, PS is associated with several adverse reactions in clinical applications, such as systemic hypotension, pulmonary hypertension, and allergic reactions [11,12]. Despite extensive exploration of its mechanisms, the underlying causes of PS-related adverse reactions remain unclear [13–16]. Moreover, the lack of alternative heparin antagonists has led to the cautious use of PS since 1939. Reports indicate that mild adverse reactions to PS can be as high as 16%, and unknown pathological effects may increase postoperative morbidity and mortality [17]. The unclear mechanisms of PS-related adverse reactions also pose challenges for the development of new heparin antagonists. Literature suggests that adverse reactions caused by PS may result from multiple contributing factors. Clinically, PS is also used as an excipient for long-acting insulin to prolong its effect and improve adherence to insulin therapy. Due to its strong immunogenicity, patients with diabetes who use this formulation long-term often develop antibodies against PS, making them more susceptible to adverse reactions during surgeries requiring extensive use of PS [18].

In contrast to PS, the synthetic peptide R15 developed in our laboratory offers several potential advantages as a heparin antagonist, warranting further research and development [19,20]. This potential UFH antagonist is a linear peptide composed of 15 arginine residues (Arg-Arg-Arg-Arg-Arg[5]-Arg-Arg-Arg-Arg-Arg[10]-Arg-Arg-Arg-Arg-Arg[15]), with a molecular weight of 2360.85 and a purity exceeding 99%. Preliminary studies conducted by our laboratory have assessed the pharmacodynamics and safety of R15, revealing that it possesses a heparin antagonistic ability similar to that of PS, while avoiding immunogenicity and cross-reactivity [19,20]. Additionally, R15 is easy to synthesize and allows for precise quality standards. Our research has shown that R15 is prone to degradation in plasma, and its degradation products may also

exhibit UFH antagonistic properties. Consequently, we have developed a stable total UFH antagonistic activity detection method with a low quantification limit, which has been validated according to the Chinese Pharmacopoeia. Using this assay, we subsequently characterized the pharmacokinetics of R15 in rats; the resulting data showed that R15 persisted in circulation far longer than the rapidly eliminated PS.

## Materials and methods

### Regarding drug dosage and concentration units in this article

Since the toxicity of both PS and R15 correlates with their heparin antagonism capacity (i.e., potency), the units used for PS and R15 in most experiments are $U \cdot mL^{-1}$ or $U \cdot kg^{-1}$. Employing activity units (U) allows for a more scientifically rigorous comparison of the pharmacokinetic parameters of PS and R15 under equivalent activity levels (i.e., equivalent heparin antagonism capacity).

The specific activity of PS is 150 $U \cdot mg^{-1}$, meaning that each milligram of PS can exactly neutralize 150 units of UFH. The specific activity of R15 is 170 $U \cdot mg^{-1}$, meaning that each milligram of R15 can exactly neutralize 170 units of UFH. Therefore, expressing the concentration of PS as 1 $U \cdot mL^{-1}$ denotes that each milliliter of the medium contains sufficient PS to neutralize 1 unit of UFH, and the same applies to R15.

Conversion formulas from $U \cdot mL^{-1}$ to $\mu g \cdot mL^{-1}$:

$$PS : X\left(U \cdot mL^{-1}\right) = X \times \frac{1000}{150}(\mu g \cdot mL^{-1})$$

(1)

$$R15 : X\left(U \cdot mL^{-1}\right) = X \times \frac{1000}{170}(\mu g \cdot mL^{-1})$$

(2)

### Chemicals, reagents and animals

R15 was obtained from Beijing SciLight Biotechnology Co., Ltd., China (Catalog No. C15393210). PS was sourced from Sigma, USA (Catalog No. SLBX6075). UFH was prepared by Jiangsu Wanbang, China (Batch No. 51606102). Glycerol (Batch No. 20150728) and glacial acetic acid (Batch No. 20181226) were purchased from China National Pharmaceutical Group, China. Methanol was acquired from Thermo Fisher, USA (Catalog No. 150549). Sodium citrate was obtained from Beijing Chemical Reagent, China (Catalog No. XW00680422); heparin anti-FXa kit (reagents inside: Factor Xa, Factor Xa substrate, AT, and buffer) were purchased from Biophen BioMed Company (Neuville-sur-Oise, France).

Thirty-two Wistar rats (approximately 250 g; equally divided by sex) were used in this pharmaceutical experiment. Twenty-four animals were utilized for the R15 pharmacokinetic study, while 8 were employed for the PS study. The animals were kept under standard laboratory conditions, with a relative humidity of 40−70%, a temperature range of 20−26°C, and a 12-hour light/dark cycle. All procedures for handling animals complied with the National Laboratory Animal Health Guidelines, and the study was approved and reviewed by the Institute of Military Medicine, Radiological Medicine Research Institute in Beijing, China (IACUC-DWZX-2020−503). Anesthesia was induced with 5% isoflurane and maintained at 2% throughout carotid artery cannulation. Postoperatively, animals were monitored daily to assess recovery and acclimatization. At study completion, rats were re-anesthetized with 5% isoflurane and euthanized by gradual-fill carbon dioxide in a sealed chamber (target 30–70% $CO_2$) until respiratory arrest; death was confirmed by absence of cardiac activity. All procedures adhered to the 3Rs (Replacement, Reduction, Refinement), with pre-specified humane endpoints and scheduled pain assessments; animals meeting endpoint criteria were promptly euthanized.

## Preparation of stock solutions, calibration standards (CS) and quality control (QC) samples

Preparation of Stock Solutions: Appropriate amounts of PS and R15 powders were accurately weighed and dissolved in deionized water to obtain stock solutions at 10 mg·mL$^{-1}$, respectively. This solution was stored in a refrigerator at 4 °C and replaced regularly.

UFH was accurately measured and diluted with deionized water to a final concentration of 1000 U·mL$^{-1}$. This solution was also stored at 4°C and replaced regularly. Prior to use, UFH was further diluted to fixed concentration (0.14 U·mL$^{-1}$ for PS and 0.15 U·mL$^{-1}$ for R15) using R4 buffer.

Calibration Standards: The CS of PS was achieved by the dilution of PS stock solution from 10 mg·mL$^{-1}$ to 0.1 µg·mL$^{-1}$, 0.2 µg·mL$^{-1}$, 0.4 µg·mL$^{-1}$, 0.8 µg·mL$^{-1}$, 1.0 µg·mL$^{-1}$, 1.2 µg·mL$^{-1}$, 1.4 µg·mL$^{-1}$, 1.6 µg·mL$^{-1}$, 1.8 µg·mL$^{-1}$ and 2.0 µg·mL$^{-1}$ using EDTA-containing sodium citrate anticoagulated rat plasma (EDTA: 0.5 mg·mL$^{-1}$, final concentration). This is the working solution of CS, wherein 0.1 µg·mL$^{-1}$, 0.2 µg·mL$^{-1}$ and 2 µg·mL$^{-1}$ are set as the anchor points on the CS (anchor points: participate in CS fitting, but not in RE% calculation). R15 stock solution was diluted with same EDTA-containing rat plasma to prepare a series of working solutions at the following concentrations: 0.1 µg·mL$^{-1}$, 0.2 µg·mL$^{-1}$, 0.4 µg·mL$^{-1}$, 0.6 µg·mL$^{-1}$, 0.8 µg·mL$^{-1}$, 1.0 µg·mL$^{-1}$, 1.2 µg·mL$^{-1}$, 1.4 µg·mL$^{-1}$, 1.6 µg·mL$^{-1}$ and 1.8 µg·mL$^{-1}$. Among these, 0.1 µg·mL$^{-1}$, 0.2 µg·mL$^{-1}$ and 1.8 µg·mL$^{-1}$ served as anchor points on the calibration curve. EDTA can inhibit the cleavage of C-terminal arginine residues by metallocarboxypeptidase [21], allowing R15 to remain stabilized in plasma. **Unless otherwise specified, all plasma samples containing PS or R15 were supplemented with EDTA to prevent degradation; any deviations are explicitly noted.** The unit conversion for the calibration standards of PS and R15 is as follows: PS: 0.4 µg·mL$^{-1}$ to 1.8 µg·mL$^{-1}$, corresponding to 0.060 U·mL$^{-1}$ to 0.270 U·mL$^{-1}$ when converted to activity units. R15: 0.4 µg·mL$^{-1}$ to 1.6 µg·mL$^{-1}$, corresponding to 0.068 U·mL$^{-1}$ to 0.272 U·mL$^{-1}$ in activity units.

Quality Control Samples: 0.7 µg·mL$^{-1}$, 1.1 µg·mL$^{-1}$ and 1.5 µg·mL$^{-1}$ of PS were prepared by the dilution of stock solution of PS using ironized water. R15 stock solution was diluted with rat plasma to prepare quality control samples at three concentrations: 0.7 µg·mL$^{-1}$, 1.1 µg·mL$^{-1}$ and 1.5 µg·mL$^{-1}$, representing low, medium, and high QC levels, respectively.

## Sample preparation

Processing of Plasma Samples: 100 µL of the unknown plasma sample was added to 500 µL of 75% methanol (methanol: water = 75:25, *v/v*) and vortex-mixed for 5 seconds. The mixture was then centrifuged at 12,000 r/min for 10 minutes at 25°C. The supernatant (500 µL) was collected and dried under nitrogen at 60°C. Subsequently, UFH (0.14 U·mL$^{-1}$ for PS and 0.15 U·mL$^{-1}$ for R15; 120 µL) was added, and the mixture was vigorously shaken at room temperature for 10 minutes. The residual UFH content was measured using the Heparin Anti-FXa assay kit.

The detailed procedure is as follows: Accurately pipette 40 µL of the sample into a 96-well plate. Place the plate in a microplate shaker. Using a multichannel pipette, add 40 µL of reagent R1 and mix by shaking. Incubate at 37 °C for exactly 3 minutes (start timing upon addition of R1; at 50 seconds, begin shaking with the microplate shaker for 1 minute, then place the plate in the incubator). Remove the 96-well plate from the incubator and place it on the open microplate shaker. Using a multichannel pipette, add 40 µL of reagent R2 to each well and incubate at 37 °C for 6 minutes (after adding R2, at 3 minutes and 50 seconds, open the microplate shaker and shake for 4 minutes, then return the plate to the incubator). Remove the 96-well plate from the incubator and place it on the open microplate shaker. Quickly add 40 µL of reagent R3 using a multichannel pipette and shake until 9 minutes and 10 seconds. At 9 minutes and 30 seconds, add 80 µL of 20% acetic acid to each well to terminate the reaction. Measure the absorbance at 405 nm using a microplate reader (Multiskan FC, Thermo Fisher US).

## Methodology

A full validation of this bioanalytical method was carried out according to the Chinese Pharmacopoeia [22]. All the preparation of blank plasma of rats were shown as follows: the whole blood of rats were collected from the heart and

anticoagulated using trisodium citrate dihydrate (9:1; *v/v*). The plasma was obtained by the centrifugating at 8000 r·min$^{-1}$ for 10 min at 4°C. The determination of samples for method validation please see "Sample Preparation".

## Linearity and range

Ten concentration levels of CS of PS or R15 were applied to plot the linear relationship between absorbance value at 405 nm and analyte concentration by nonlinear regression (Sigmoidal, 4PL). The Equation was as follows:

$$Y(O.D.) = A2 + (A1 - A2)/(1 + \left(\frac{X}{X_0}\right)^P)$$

(3)

Ten (7 valid) concentrations, in this method, were used to establish the standard curve, and three anchor points were used to assist the curve fitting. Each of the recalculated concentration of the calibration standard should be within the range of ± 20% of the nominal concentration (the lower limit and upper limit of quantification were ± 25%). The concentration range between the lower limit and upper limit of is the valid concentration range of the standard curve. Anchor point correction samples are standard samples outside the quantitative range, which are used to assist fitting the nonlinear regression standard curve of this method, but do not follow the above acceptance criteria.

## Selectivity

Due to the presence of unrelated substances in samples that may interfere with the analytical method establishment and validation of the analyte, necessary evaluation of the interference between matrix and samples should be required. Ten blank plasma of rat were prepared from 10 individual rats to evaluate the established method for selectivity.

The stock solution of PS and R15 were diluted in 10 rats' plasma to concentration of Lower Limit of Quantification (LLOQ; 0.4 µg·mL$^{-1}$) and Upper Limit of Quantification (ULOQ; 1.8 µg·mL$^{-1}$ for PS and 1.6 µg·mL$^{-1}$ for R15), respectively. Blank (no analytes), LLOQ and ULOQ were determined by an CS with QC samples to investigate the interference between analytes and irrelevant substance in the matrix.

## Precision and accuracy

Five concentration levels (LLOQ, and low, medium and high QC, ULOQ; three samples per level) were selected to evaluate accuracy and precision in six analytical runs on separate days. Intra- and inter-day precision was evaluated through the relative standard deviation (RSD%) of repeated measurement results, which should be less than ± 20% (LLOQ and ULOQ are less than ± 25%). Accuracy was measured from the difference between determined and nominal concentrations (relative error, RE%), which should be within ± 20% for the three QC and within ± 25% for the LLOQ and ULOQ. Samples for method validation should be frozen and treated before determination as the process of real samples. In addition, the intra- and inter-day total error of the method (the sum of absolute intra- and inter-day of RE% and RSD%) should not exceed 30% (lower limit and upper limit of quantification are 40%).

## Stability

Two concentration levels (low and high QC) were evaluated for validation of storage stability and freeze/thaw stability. Two concentration levels of PS and R15 (low and high QC) in blank rat plasma were prepared (n = 6), respectively. Samples in rats' plasma were determined after storage at room temperature (25°C ± 1°C) for 30 min. Samples in rats' plasma were under three freeze (−80°C)-thaw (25°C) cycles. Stock solution of PS and R15 were stored at 4°C for a week. All the samples mentioned above were processed by calibration curves with adequate QC samples diluted from newly prepared stock solution of PS and R15, followed by procedure of sample determination (see "Sample Preparation"), respectively.

### Dilution effects

Samples in rats' plasma (n = 5) containing PS were diluted 2-, 5-, 10- and 20-fold to 1 µg·mL$^{-1}$ with blank plasma of rat, and measured using a freshly prepared calibration curve. Acceptance criteria for dilution effects were less than 20% for the accuracy (RE%) and precision (RSD%) of the diluted samples.

Quality control samples were diluted 2-fold and 100-fold to assess whether the measured values corresponded with theoretical values. Quality control samples at concentrations of 0.7 ug·mL$^{-1}$ and 1.5 µg·mL$^{-1}$ were prepared and diluted 2-fold and 100-fold using blank plasma of rat. Five samples were prepared for each concentration. These were compared with the calibration curve prepared from freshly collected plasma on the same day.

### Pharmacokinetic study

**Animal experiments.** Eight Wistar rats (approximately 250 g; equally divided by sex and acclimatized for 1 week in the animal facility) were used for profiling pharmacokinetics of PS. Each rat was cannulated in carotid artery to collect blood the day before the experiment. PS was injected through the tail vein and blood samples were collected at 0 min, 1 min, 3 min, 5 min, 7 min, 9 min, 11 min, 15 min, 20 min, 30 min, 60 min and 120 min in a tube containing 4% sodium citrate. Plasma was achieved by centrifugation at 8000 r/min for 10 min, and stored in the refrigerator at −80°C.

Twenty-four Wistar rats (approximately 250 g; equally divided by sex and acclimatized for 1 week in the animal facility) were divided into three groups: high-dose group (2700 U·kg$^{-1}$), medium-dose group (900 U·kg$^{-1}$), and low-dose group (300 U·kg$^{-1}$), with 8 rats in each group. Prior to the experiment, each rat underwent carotid artery cannulation. The tail vein was used to administer the respective doses of R15, and blood samples were collected via the carotid artery cannula. Blood samples from the high-dose group were collected at the following time points post-administration: 0 min, 1 min, 5 min, 15 min, 30 min, 60 min, 120 min, 240 min, 360 min, 480 min, 600 min, 720 min, and 840 min. The medium-dose group had blood collected at: 0 min, 1 min, 5 min, 15 min, 30 min, 60 min, 120 min, 240 min, 360 min, 480 min, 600 min, and 720 min. The low-dose group had blood collected at: 0 min, 1 min, 5 min, 15 min, 30 min, 60 min, 120 min, 240 min, 360 min, 480 min, and 600 min. All the blood samples were anticoagulated using 4% sodium citrate. Plasma was obtained by centrifugation at 8000 r/min for 10 min and stored at −80°C.

### Sample analysis

During the analysis of actual samples, a standard curve and a series of calibration standards were established for each analytical batch to determine the concentration of PS and R15 in plasma. The concentrations were plotted on the X-axis, and the data were fitted using Equation (3).

Weighted least squares regression was used for the calculations. The Optical Density (OD) values of the unknown biological samples were substituted into the equation to determine the concentration of PS and R15, respectively.

### Statistical analysis

Pharmacokinetic parameters of PS were calculated by WinNonlin 6.4, and all data were processed by Microsoft Excel 2013 and GraphPad Prism 8.3.

## Results

### Method validation

**Linearity and range.** The reliable concentration range of PS and R15 detected by this method is 0.4 µg·mL$^{-1}$ ~ 1.8 µg·mL$^{-1}$ and 0.4 µg·mL$^{-1}$ ~ 1.6 µg·mL$^{-1}$, respectively. Tables 1 and 2 lists the six parameters of the calibration curve in six individual analysis batches. The anchor points (red points) were set to aid the curve fitting (Figs 1 and 2). The assay exhibited a reliable, linear concentration range of 0.4 µg·mL$^{-1}$ to 1.8 µg·mL$^{-1}$ (equivalent to 0.06 U·mL$^{-1}$ to 0.27 U·mL$^{-1}$) for

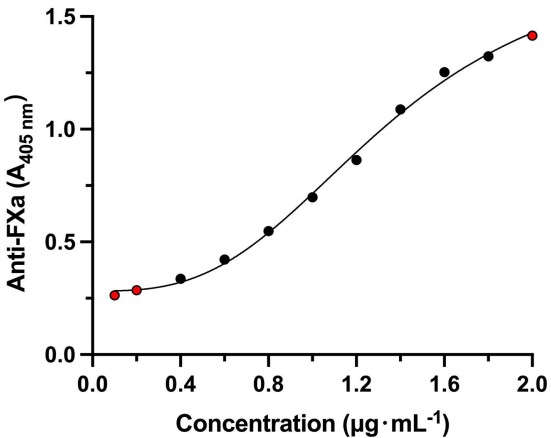

**Table 1. The standard curve parameters of PS for 6 runs in blank plasma.**

| Analysis batch | A1 | A2 | x0 | p | R2 |
|---|---|---|---|---|---|
| 1 | 0.3100 | 2.6146 | 1.8293 | 2.1627 | 0.9907 |
| 2 | 0.2865 | 1.7236 | 1.3760 | 2.6603 | 0.9970 |
| 3 | 0.3783 | 3.1042 | 2.1920 | 1.9445 | 0.9922 |
| 4 | 0.3563 | 1.9042 | 1.1856 | 2.6553 | 0.9961 |
| 5 | 0.3221 | 2.0999 | 1.5521 | 2.3894 | 0.9944 |
| 6 | 0.2837 | 3.4871 | 2.5559 | 1.8531 | 0.9955 |

**Table 2. The standard curve parameters of R15 for 6 runs in blank plasma.**

| Analysis batch | A1 | A2 | x0 | p | R2 |
|---|---|---|---|---|---|
| 1 | 0.3265 | 1.6380 | 0.8948 | 2.2590 | 0.9976 |
| 2 | 0.3190 | 1.6690 | 0.9557 | 2.0500 | 0.9951 |
| 3 | 0.3923 | 1.5600 | 0.8738 | 2.2730 | 0.9977 |
| 4 | 0.4085 | 1.7560 | 0.9101 | 1.8660 | 0.9980 |
| 5 | 0.3302 | 1.6350 | 0.8557 | 2.1120 | 0.9982 |
| 6 | 0.3801 | 1.7650 | 0.8809 | 1.8610 | 0.9988 |

**Fig 1. Standard curve of PS in blank plasma.** Red point is the anchor point.

PS, and 0.4 µg·mL$^{-1}$ to 1.6 µg·mL$^{-1}$ (equivalent to 0.068 U·mL$^{-1}$ to 0.272 U·mL$^{-1}$) for R15, with a lower limit of quantitation of 0.4 µg·mL$^{-1}$.

## Selectivity

Ten individual blank plasma samples were achieved from 10 rats for both PS and R15. The PS or R15 in plasma from 10 different sources were evaluated for the possible influences due to different matrices. Tables 3 and 4 showed that PS and R15 determined by this method were not affected by different matrices of rat plasma, indicating that PS and R15 exhibited good selectivity in rat plasma, and the source of plasma does not affect the concentrations of PS and R15.

## Precision and accuracy

The PS or R15 stock solutions were diluted with blank rat plasma to 0.4 µg·mL⁻¹, 0.7 µg·mL⁻¹, 1.1 µg·mL⁻¹, 1.5 µg·mL⁻¹ and 1.8 µg·mL⁻¹ (for PS)/ 1.6 µg·mL⁻¹ (for R15), respectively. A total of 6 batches of calibration curves and QC samples were prepared for precision and accuracy evaluation. For details of results, please see Tables 5 and 6. Precision and accuracy were within the acceptable limits of ± 25% for LLOQ and ULOQ, and ± 20% for the three QC concentration levels. The total intra- and inter-day precision and accuracy meet the requirements of method validation.

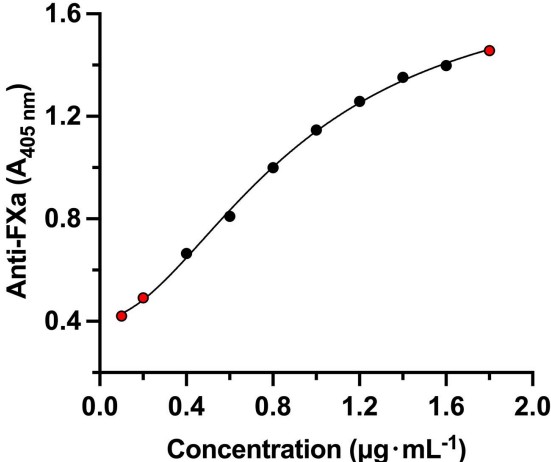

**Fig 2. Standard curve of R15 in blank rat plasma.** Red point is the anchor point.

**Table 3. The selectivity of PS (n = 10).**

| NC (µg·mL⁻¹) | Type | 1# | 2# | 3# | 4# | 5# | 6# | 7# | 8# | 9# | 10# |
|---|---|---|---|---|---|---|---|---|---|---|---|
| 0 | MC (µg·mL⁻¹) | ND | ND | ND | ND | ND | ND | ND | ND | ND | ND |
| 0.4 | | 0.43 | 0.46 | 0.45 | 0.46 | 0.41 | 0.47 | 0.47 | 0.47 | 0.42 | 0.42 |
| 1.8 | | 1.82 | 1.81 | 1.82 | 1.77 | 1.84 | 1.81 | 1.81 | 1.75 | 1.79 | 1.84 |
| 0.4 | RE (%) | 8.63 | 13.85 | 11.54 | 16.15 | 2.01 | 16.69 | 16.44 | 17.62 | 5.17 | 4.97 |
| 1.8 | | 0.89 | 0.80 | 0.90 | −1.55 | 2.04 | 0.35 | 0.39 | −2.93 | −0.39 | 2.15 |

NC: nominal concentration; ND: not detected; MC: measured concentration.

**Table 4. The selectivity of R15 (n = 10).**

| NC (µg·mL⁻¹) | Type | 1# | 2# | 3# | 4# | 5# | 6# | 7# | 8# | 9# | 10# |
|---|---|---|---|---|---|---|---|---|---|---|---|
| 0 | MC (µg·mL⁻¹) | ND | ND | ND | ND | ND | ND | ND | ND | ND | ND |
| 0.4 | | 0.40 | 0.37 | 0.36 | 0.37 | 0.37 | 0.38 | 0.36 | 0.40 | 0.35 | 0.36 |
| 1.6 | | 1.52 | 1.52 | 1.50 | 1.49 | 1.51 | 1.50 | 1.49 | 1.51 | 1.49 | 1.45 |
| 0.4 | RE (%) | 0.64 | −8.47 | −9.69 | −9.09 | −9.06 | −5.61 | −9.89 | 1.15 | −14.43 | −12.19 |
| 1.6 | | −5.20 | −5.44 | −6.66 | −7.15 | −5.67 | −7.00 | −7.39 | −6.13 | −7.63 | −10.26 |

NC: nominal concentration; ND: not detected; MC: measured concentration.

## Stability

Stability of PS or R15 in plasma at room temperature for 30 min and after three freeze-thaw cycles was investigated, as well as the stability of the stock solution at 4°C for a week. The results are all within the acceptance criteria (RE% < 20%) and summarized in Tables 7 and 8.

## Dilution effects

The PS in plasma at concentration of 2 µg·mL⁻¹, 5 µg·mL⁻¹, 10 µg·mL⁻¹ and 20 µg·mL⁻¹ (n = 5) were diluted by 2-, 5-, 10- and 20-fold times to 1 µg·mL⁻¹, respectively. The results were all within the acceptance criteria (RE% < 20%, RSD% < 20%) and summarized in Table 9. Samples were diluted 2-fold and 100-fold to investigate whether the measured values corresponded with the theoretical values. Samples were prepared and diluted 2-fold and 100-fold using rat blank plasma to reach concentrations of 0.7 µg·mL⁻¹ and 1.5 µg·mL⁻¹. Five samples were prepared for each concentration. The results were compared with the calibration curve prepared on the same day. The experimental results, shown in Table 10, indicate that there is no dilution effect on R15 within the dilution range of 2-fold to 100-fold.

**Table 5. Precision and accuracy of method to determine PS.**

| Batch number | Concentration (µg·mL⁻¹) | | | | |
|---|---|---|---|---|---|
| | 1.8 | 1.5 | 1.1 | 0.7 | 0.4 |
| 1 | 1.71 | 1.42 | 1.05 | 0.70 | 0.38 |
| | 1.77 | 1.47 | 1.10 | 0.73 | 0.42 |
| | 1.69 | 1.42 | 1.08 | 0.71 | 0.38 |
| 2 | 1.77 | 1.59 | 1.11 | 0.71 | 0.38 |
| | 1.73 | 1.52 | 1.12 | 0.76 | 0.42 |
| | 1.81 | 1.65 | 1.18 | 0.70 | 0.42 |
| 3 | 1.72 | 1.44 | 1.04 | 0.68 | 0.37 |
| | 1.77 | 1.45 | 1.05 | 0.71 | 0.40 |
| | 1.78 | 1.44 | 1.08 | 0.71 | 0.40 |
| 4 | 1.75 | 1.44 | 1.03 | 0.68 | 0.41 |
| | 1.74 | 1.49 | 1.06 | 0.74 | 0.40 |
| | 1.81 | 1.47 | 1.08 | 0.70 | 0.37 |
| 5 | 1.70 | 1.48 | 1.04 | 0.67 | 0.39 |
| | 1.68 | 1.49 | 1.06 | 0.66 | 0.39 |
| | 1.66 | 1.42 | 1.02 | 0.65 | 0.39 |
| 6 | 1.69 | 1.50 | 1.11 | 0.73 | 0.41 |
| | 1.66 | 1.47 | 1.08 | 0.70 | 0.41 |
| | 1.69 | 1.50 | 1.10 | 0.70 | 0.39 |
| n | 18 | 18 | 18 | 18 | 18 |
| Mean | 1.73 | 1.48 | 1.08 | 0.70 | 0.40 |
| Inter-batch SD | 0.07 | 0.10 | 0.06 | 0.04 | 0.01 |
| Intra-batch SD | 0.03 | 0.03 | 0.03 | 0.02 | 0.02 |
| Inter-batch RSD (%) | 4.3 | 6.5 | 5.8 | 5.6 | 2.7 |
| Intra-batch RSD (%) | 1.9 | 2.3 | 2.4 | 3.0 | 4.7 |
| RE (%) | −3.95 | −1.21 | −2.04 | 0.37 | −1.25 |
| Total inter-batch error (%) | 8.19 | 7.73 | 7.81 | 5.97 | 3.98 |
| Total intra-batch error (%) | 5.81 | 3.52 | 4.40 | 3.35 | 5.95 |

Table 6. Precision and accuracy for the R15 in Wistar plasma.

| Batch number | Concentration (µg·mL⁻¹) | | | | |
|---|---|---|---|---|---|
| | 0.4 | 0.7 | 1.1 | 1.5 | 1.6 |
| 1 | 0.46 | 0.80 | 1.16 | 1.68 | 1.67 |
| | 0.49 | 0.78 | 1.14 | 1.62 | 1.75 |
| | 0.46 | 0.71 | 1.27 | 1.69 | 1.69 |
| 2 | 0.40 | 0.64 | 1.11 | 1.35 | 1.53 |
| | 0.39 | 0.65 | 1.06 | 1.35 | 1.58 |
| | 0.40 | 0.61 | 1.08 | 1.30 | 1.52 |
| 3 | 0.42 | 0.73 | 1.10 | 1.54 | 1.54 |
| | 0.45 | 0.71 | 1.19 | 1.58 | 1.53 |
| | 0.43 | 0.65 | 1.11 | 1.51 | 1.51 |
| 4 | 0.43 | 0.72 | 1.20 | 1.49 | 1.47 |
| | 0.40 | 0.70 | 1.19 | 1.42 | 1.45 |
| | 0.40 | 0.70 | 1.12 | 1.42 | 1.43 |
| 5 | 0.42 | 0.74 | 1.04 | 1.23 | 1.31 |
| | 0.43 | 0.71 | 1.09 | 1.23 | 1.32 |
| | 0.40 | 0.67 | 1.08 | 1.24 | 1.29 |
| 6 | 0.36 | 0.67 | 1.17 | 1.55 | 1.61 |
| | 0.34 | 0.64 | 1.22 | 1.56 | 1.54 |
| | 0.34 | 0.64 | 1.16 | 1.53 | 1.62 |
| n | 18 | 18 | 18 | 18 | 18 |
| Mean | 0.41 | 0.69 | 1.14 | 1.46 | 1.52 |
| Inter-batch SD | 0.07 | 0.08 | 0.09 | 0.27 | 0.23 |
| Intra-batch SD | 0.01 | 0.03 | 0.04 | 0.03 | 0.03 |
| Inter-batch RSD (%) | 17.31 | 11.6 | 7.88 | 18.7 | 15.18 |
| Intra-batch RSD (%) | 3.37 | 4.36 | 3.95 | 1.97 | 1.95 |
| RE (%) | 2.87 | −1.02 | 3.51 | −2.73 | −4.99 |
| Total inter-batch error (%) | 20.18 | 12.62 | 11.39 | 21.43 | 20.16 |
| Total intra-batch error (%) | 6.24 | 5.38 | 7.45 | 4.70 | 6.94 |

Table 7. Stability data of the PS plasma sample of different storage situation (n = 6).

| Storage Conditions | NC (µg·mL⁻¹) | Mean±SD (µg·mL⁻¹) | RSD (%) | RE (%) |
|---|---|---|---|---|
| Room temperature for 30 min concentration (µg·mL⁻¹) | 0.7 | 0.76±0.01 | 1.81 | 8.10 |
| | 1.5 | 1.57±0.09 | 5.90 | 4.44 |
| Three freeze-thaw cycles Concentration (µg·mL⁻¹) | 0.7 | 0.78±0.02 | 2.80 | 10.71 |
| | 1.5 | 1.59±0.04 | 2.72 | 5.78 |
| Stock solution for 1 weeks (4°C) | 0.7 | 0.59±0.02 | 3.22 | −15.2 |
| | 1.5 | 1.44±0.04 | 2.83 | −4.1 |

NC: nominal concentration.

## Pharmacokinetics of PS and R15 in rats

A dose of 300 U· kg⁻¹ dose of UFH is considered clinically effective [23]. Preliminary experimental data showed that 300 U ·kg⁻¹ maintains heparinization in rats for at least 4 hours [20]. Accordingly, we performed pharmacokinetic studies of PS at 300 U kg⁻¹, and of R15 at 300, 900, and 2700 U· kg⁻¹ to determine whether R15 follows linear or nonlinear pharmacokinetics.

**Table 8. Stability data of the R15 plasma sample of different storage situation (n=6).**

| Storage Conditions | NC (µg·mL⁻¹) | Mean±SD (µg·mL⁻¹) | RSD (%) | RE (%) |
|---|---|---|---|---|
| Room temperature for 30 min concentration (µg·mL⁻¹) | 0.7 | 0.70±0.02 | 2.62 | 0.24 |
| | 1.5 | 1.63±0.09 | 5.44 | 8.57 |
| Three freeze-thaw cycles Concentration (µg·mL⁻¹) | 0.7 | 0.72±0.02 | 2.83 | 3.06 |
| | 1.5 | 1.54±0.09 | 5.85 | 2.60 |
| Stock solution for 1 weeks (4°C) | 0.7 | 0.74±0.02 | 3.35 | 5.86 |
| | 1.5 | 1.71±0.08 | 4.56 | 14.16 |

NC: nominal concentration.

**Table 9. Dilution effects of varying concentrations of plasma samples of PS diluted 2-fold, 5-fold, 10-fold, 20-fold (n=5).**

| NC (µg·mL⁻¹) | Fold | Mean±SD (µg·mL⁻¹) | RSD (%) | RE (%) |
|---|---|---|---|---|
| 1 | 2 | 1.02±0.03 | 3.22 | 1.7 |
| | 5 | 0.98±0.08 | 8.3 | −2.35 |
| | 20 | 1.01±0.03 | 3.37 | 1.01 |
| | 100 | 1.02±0.03 | 3.07 | 1.99 |

NC: nominal concentration.

**Table 10. Dilution effects of R15 with 2-fold dilution to 1.5 µg·mL⁻¹ and 100-fold to 0.7 µg·mL⁻¹ (n=5).**

| NC (µg·mL⁻¹) | Mean±SD (µg·mL⁻¹) | RSD (%) | RE (%) |
|---|---|---|---|
| 0.7 | 0.73±0.02 | 2.25 | 4.53 |
| 1.5 | 1.47±0.03 | 2.00 | −2.63 |

NC: nominal concentration.

## Pharmacokinetic Parameters of PS (300 U·kg⁻¹)

The plasma PS concentrations at each time point are shown in Table 11. Rats 1#, 2#, 3# and 6# were male rats, while rats 11#, 12#, 13# and 14# were female rats. Fig 3 shows the mean drug concentration-time curve of 8 rats. Pharmacokinetic parameters of rats 3# and 14# were not included in the mean values. The pharmacokinetic experiment results of PS (300 U·kg⁻¹) in Wistar rats after single tail vein injection were obtained (Rats 3# and 14# had too few drug-time curve points and the calculation of pharmacokinetic parameters was not accurate, so it was not included in the calculation of mean pharmacokinetic parameters): The maximum plasma concentration ($C_{max}$) at 1 minute after injection was $12.67 \pm 1.96$ µg mL⁻¹. The mean half-life ($T_{1/2}$) was $3.94 \pm 3.08$ min, indicating that PS is readily metabolized or excreted in the body. The distribution ($V_d$) was $268 \pm 151$ mL·kg⁻¹, which was larger than the blood volume of rats (55−70 mL·kg⁻¹), suggesting that PS might be distributed in extracellular fluid. The Clearance (CL) was $53.65 \pm 11.49$ mL·min⁻¹·kg⁻¹, indicating that PS could be easily eliminated *in vivo*. The area under the curves (AUC) was $35.89 \pm 7.07$ min·µg·mL⁻¹ and the MRT was $2.95 \pm 0.81$ min.

The observed male-female disparity in PS pharmacokinetics in rats is most plausibly attributable to differences in systemic clearance capacity. First, in humans, PS exposure is higher in men and systemic clearance is lower than in women [24,25], suggesting that the sex difference seen in rats likely reflects an intrinsic, sex-dependent biological mechanism rather than inter-individual variation among rats. In rats, PS is cleared predominantly by the kidneys [24]. At the cellular level, PS is filtered at the glomerulus and taken up by renal proximal tubular epithelium via receptor-mediated endocytosis-principally through the megalin/cubilin complex-followed by lysosomal processing [26,27]. Renal tubular

**Table 11. Pharmacokinetic parameters of intravenous administration of PS into Wistar rats (n = 8, half male and half female).**

| Parameter (Units) | 300 U·kg$^{-1}$ | | |
|---|---|---|---|
| | Male | Female | Total |
| $T_{1/2}$ (min) | 5.81 ± 3.63 | 2.08 ± 0.40 | 3.94 ± 3.08 |
| $C_{max}$ (µg·mL$^{-1}$) | 13.79 ± 1.55 | 11.55 ± 1.86 | 12.67 ± 1.96 |
| AUC (min·µg·mL$^{-1}$) | 41.48 ± 5.05 | 30.30 ± 2.40 | 35.89 ± 7.07 |
| $V_d$ (mL·kg$^{-1}$) | 349 ± 189 | 186 ± 38 | 268 ± 151 |
| CL (mL·min$^{-1}$·kg$^{-1}$) | 44.88 ± 7.77 | 62.42 ± 6.26 | 53.65 ± 11.49 |
| MRT (min) | 3.37 ± 0.96 | 2.53 ± 0.43 | 2.95 ± 0.81 |

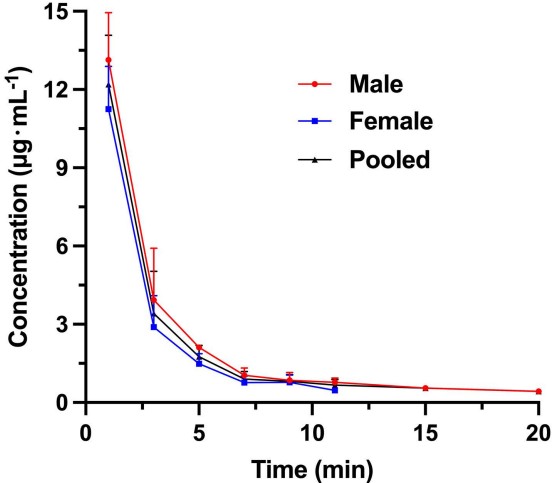

**Fig 3. The mean plasma concentration-time curve of PS after intravenous infusion administration with PS (300 U·kg$^{-1}$) to Wistar rats (male, female and pooled).**

function also exhibits sex dimorphism: compared with males, female rats have a lower fractional reabsorption of sodium and water in the proximal tubule, resulting in a greater fraction of the filtered load being delivered to downstream segments [28]. In addition, a recent rat study demonstrated higher megalin protein abundance in the renal cortex of females than males, implying greater proximal tubular endocytic/processing throughput in females; consequently, positively charged small polypeptides such as PS are removed from plasma more rapidly, and because the kidney effectively "captures" PS early, females exhibit smaller $V_d$ and MRT [28,29].

## Pharmacokinetic parameters of R15 (300 U·kg$^{-1}$)

After administering R15 (300 U·kg$^{-1}$) via the tail vein to Wistar rats, the mean plasma concentrations of R15 at various time points are presented in Fig 4. Male rats are identified as 9#, 11#, 14# and 15#, while the female rats are 18#, 19#, 20# and 30#. The pharmacokinetic results of the pharmacokinetic study following a single tail vein injection of R15 (300 U·kg$^{-1}$) in Wistar rats are summarized in Table 12. The plasma concentration at 1 min post-injection was 10.89 ± 2.43 µg·mL$^{-1}$, and the mean $T_{1/2}$ was 51.93 ± 12.38 minutes, indicating that R15 can maintain anti-UFH activity for a longer period, compared with PS. The $V_d$ was 194 ± 24 mL·kg$^{-1}$ and the CL was 2.73 ± 0.75 mL·U·min$^{-1}$·kg$^{-1}$. The AUC was 632 ± 182 min·µg·mL$^{-1}$, and the MRT was 54.95 ± 12.40 min.

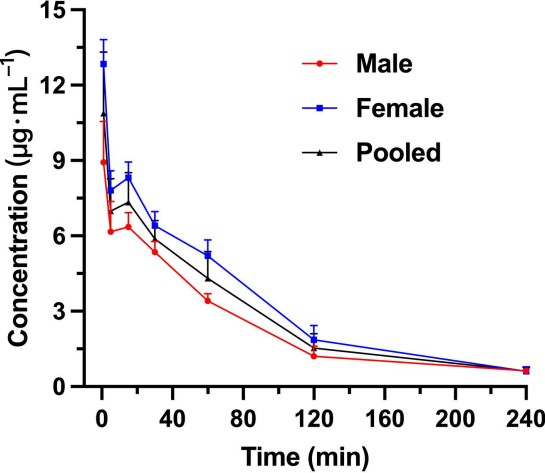

**Fig 4. The mean plasma *concentration-time curve of R15 after intravenous infusion administration* with R15 (300 U·kg⁻¹) to Wistar rats (male, female and pooled).**

**Table 12. Pharmacokinetic parameters of intravenous administration of R15 into Wistar rats (n=8, half male and half female).**

| Parameter (Units) | Male | | | Female | | | Total | | |
|---|---|---|---|---|---|---|---|---|---|
| | 300 U·kg⁻¹ | 900 U·kg⁻¹ | 2700 U·kg⁻¹ | 300 U·kg⁻¹ | 900 U·kg⁻¹ | 2700 U·kg⁻¹ | 300 U·kg⁻¹ | 900 U·kg⁻¹ | 2700 U·kg⁻¹ |
| $T_{1/2}$ (min) | 45.04±13.76 | 89.15±21.17 | 151.89±65.51 | 58.82±6.47 | 93.11±11.54 | 174.34±47.94 | 51.93±12.38 | 91.13±15.92 | 163.12±54.48 |
| $C_{max}$ (µg·mL⁻¹) | 8.93±1.63 | 11.46±1.41 | 12.94±1.55 | 12.84±0.97 | 11.62±1.02 | 10.37±3.68 | 10.89±2.43 | 11.54±1.14 | 11.65±2.95 |
| AUC (min·µg·mL⁻¹) | 491±115 | 999±209 | 1241±521 | 774±106 | 1063±166 | 1399±249 | 632±182 | 1030±178 | 1320±387 |
| $V_d$ (mL·kg⁻¹) | 206±26 | 616±56 | 2658±1356 | 183±17 | 599±69 | 2579±655 | 194±24 | 608±59 | 2618±987 |
| CL (mL·min⁻¹·kg⁻¹) | 3.30±0.59 | 4.95±0.98 | 11.97±3.37 | 2.17±0.31 | 4.50±0.67 | 10.35±1.63 | 2.73±0.75 | 4.72±0.81 | 11.16±2.60 |
| MRT (min) | 46.73±12.71 | 94.00±28.41 | 181.83±63.16 | 63.16±4.15 | 94.80±16.80 | 205.91±41.44 | 54.95±12.40 | 94.40±21.61 | 193.87±51.11 |

Next, we compared the PK parameters of PS and R15 at the same dose of 300 U·kg⁻¹ (Table 13). Across females, males, and pooled analyses, R15 showed statistically significant differences from PS in $T_{1/2}$, AUC, CL, and MRT, except for male AUC and MRT and female CL. The most likely explanation is the small sample size and inter-individual variability leading to non-normal distributions. From the pooled data, compared with PS, R15 exhibits higher systemic exposure and residence (AUC and MRT), a longer $T_{1/2}$, and a lower CL. The $V_d$ values were not significantly different, indicating that the two compounds occupy similar initial distribution volumes; consequently, the observed differences in systemic exposure are driven primarily by variations in clearance rather than distribution. Consistent with the raw data, the $T_{1/2}$ and MRT of R15 are approximately 13-fold and 19-fold those of PS, respectively; CL is about 1/20 that of PS; AUC is about 18-fold that of PS; and both $V_d$ and $C_{max}$ are slightly lower for R15 (Tables 11 and 12). These differences are mainly attributable to the fact that active metabolites of R15 retain pharmacological activity after degradation of the parent peptide, effectively mitigating heparin rebound caused by the rapid degradation of PS.

**Table 13. Statistical comparison of pharmacokinetic parameters between PS and R15 (300 U·kg⁻¹).**

| Parameter (Units) | PS 300 U·kg⁻¹ | | | R15 300 U·kg⁻¹ | | | Normality test | | | Statistical analysis | | |
|---|---|---|---|---|---|---|---|---|---|---|---|---|
| | Male | Female | Total | Male | Female | Total | Male | Female | Pooled | Male | Female | Pooled |
| $T_{1/2}$ (min) | 5.81±3.63 | 2.08±0.40 | 3.94±3.08 | 45.04±13.76 | 58.82±6.47 | 51.93±12.38 | Yes | Yes | No | ** | **** | *** |
| $C_{max}$ (µg·mL⁻¹) | 13.79±1.55 | 11.55±1.86 | 12.67±1.96 | 8.93±1.63 | 12.84±0.97 | 10.89±2.43 | Yes | Yes | Yes | * | No | No |
| AUC (min·µg·mL⁻¹) | 41.48±5.05 | 30.30±2.40 | 35.89±7.07 | 491±115 | 774±106 | 632±182 | No | Yes | Yes | No p=0.057 | **** | **** |
| $V_d$ (mL·kg⁻¹) | 349±189 | 186±38 | 268±151 | 206±26 | 183±17 | 194±24 | Yes | Yes | Yes | No | No | No |
| CL (mL·min⁻¹·kg⁻¹) | 44.88±7.77 | 62.42±6.26 | 53.65±11.49 | 3.30±0.59 | 2.17±0.31 | 2.73±0.75 | Yes | No | Yes | *** | No p=0.057 | **** |
| MRT (min) | 3.37±0.96 | 2.53±0.43 | 2.95±0.81 | 46.73±12.71 | 63.16±4.15 | 54.95±12.40 | No | Yes | No | No p=0.057 | **** | *** |

For normally distributed data, parametric analysis was performed using an unpaired two-sample t-test; for non-normally distributed data, non-parametric analysis was conducted with the Mann–Whitney U test. $*p < 0.05$, $** p < 0.01$, $*** p < 0.001$, $**** p < 0.0001$.

### Pharmacokinetic parameters of R15 (900 U·kg⁻¹)

After administering R15 (900 U·kg⁻¹) via the tail vein to Wistar rats, the mean plasma concentrations of R15 at various time points are presented in Fig 5. The male rats are identified as 2#, 3#, 4#, and 12#, while the female rats are 16#, 23#, 24#, and 26#. The results of the pharmacokinetic study following a single tail vein injection of R15 (900 U·kg⁻¹) in Wistar rats are summarized in Table 12. The plasma concentration at 1 min post-injection was 11.54 ± 1.14 µg·mL⁻¹; the mean $T_{1/2}$ was 91.13 ± 15.92 min; the $V_d$ was 608 ± 59 mL·kg⁻¹; and the CL was 4.72 ± 0.81 mL·min⁻¹·kg⁻¹. The AUC was 1030 ± 178 min·µg·mL⁻¹ and the MRT was 94.40 ± 21.61 min.

### Pharmacokinetic parameters of R15 (2700 U·kg⁻¹)

After administering R15 (2700 U·kg⁻¹) via the tail vein to Wistar rats, the mean plasma concentrations of R15 at various time points are shown in Fig 6. Male rats are identified as 1#, 5#, 8#, and 13#, while the female rats are 21#, 22#, 27#, and 28#. The results of the pharmacokinetic study following a single tail vein injection of R15 (2700 U·kg⁻¹) in Wistar rats are as follows (Table 12): the plasma concentration at 1 minute post-injection was 11.65 ± 2.95 µg·mL⁻¹; the average $T_{1/2}$ was 163.12 ± 54.48 minutes; the $V_d$ was 2618 ± 987 mL·kg⁻¹ and the CL was 11.16 ± 2.60 mL·min⁻¹·kg⁻¹. The area AUC was 1320 ± 387 min·µg·mL⁻¹ and the MRT was 193.87 ± 51.11 min.

## Discussion

In this study, we developed an *in vivo* assay to quantify the anti-heparin activity of PS and R15 and conducted subsequent pharmacokinetic studies in rats. The results indicate that R15 is a potentially safer alternative to PS for heparin neutralization.

Although PS has remained the only FDA-approved agent for neutralizing UFH in systemic anticoagulation for nearly nine decades, its well-recognized adverse reactions have driven sustained efforts to develop safer and more effective alternatives. In clinical use, PS can elicit common but seldom life-threatening episodes of transient hypotension/bradycardia [30]; hypersensitivity or anaphylactoid reactions that markedly increase mortality [31,32]; catastrophic pulmonary hypertension requiring re-establishment of cardiopulmonary bypass, which is acutely life-threatening [33,34]; non-cardiogenic pulmonary edema and new-onset postoperative atrial fibrillation (POAF) [33]; as well as bleeding due

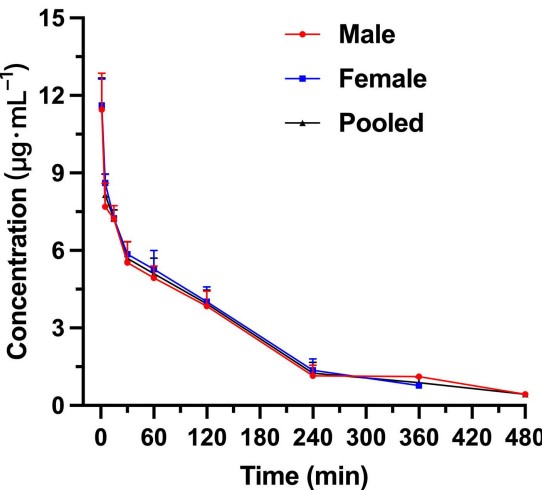

**Fig 5. The mean plasma concentration-time curve of R15 after intravenous infusion administration with R15 (900 U·kg⁻¹) to Wistar rats (male, female and pooled).**

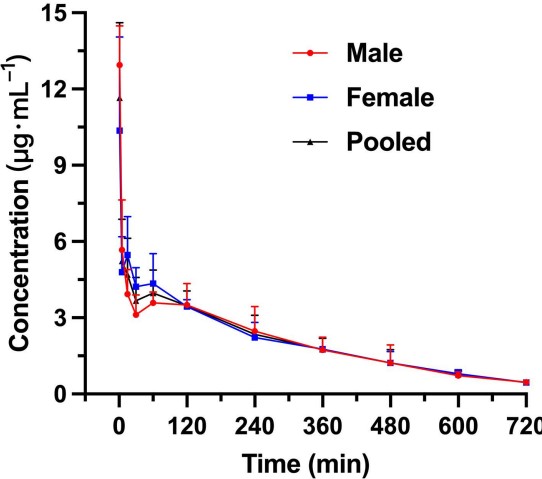

**Fig 6. The mean plasma concentration-time curve of R15 after intravenous infusion administration with R15 (2700 U·kg⁻¹) to Wistar rats (male, female and pooled).**

to heparin rebound [35]. Though infrequent, pulmonary hypertension and allergic reactions carry the highest fatality risk among adverse events linked to PS. Patients receiving protamine-containing insulin preparations constitute a high-risk group for protamine allergy [6].

PS and UFH-PS complexes activate complement [36,37]. Complement fragments can accumulate in the pulmonary microcirculation, up-regulate endothelial adhesion molecules, and recruit neutrophils and monocytes, forming inflammatory microthrombi [38,39]. Our study shows that, under UFH excess, the UFH-PS complex triggers explosive complement activation that is 8.28-fold higher than that induced by the UFH–R15 complex under the same UFH-excess conditions [20].

In clinical practice, PS is typically administered at 1.2 times the anti-heparin activity of circulating UFH to achieve complete neutralization [40,41]. However, owing to the marked difference in clearance between UFH and PS ($T_{1/2}$: PS 7.4 min vs. UFH 60–120 min) [42,43], degradation of UFH–PS complexes can regenerate free UFH, leading to coagulopathy known as heparin rebound. Moreover, because PS is cleared more rapidly than UFH, the UFH:PS stoichiometric ratio drifts toward UFH excess, a condition under which our previous study observed 'explosive complement activation' [20].

Unlike other *in vivo* assays for PS or PS analogues, the present method targets anti-heparin activity to determine PS or R15 in rat plasma. This approach quantifies the anti-heparin capacity of both parent compounds and their metabolites, which also retain anti-heparin activity [44–46]. Pharmacokinetic analysis revealed that R15 exhibits a half-life far longer than that of PS (3.94 min *vs.* 51.93 min), indicating that active metabolites of R15 prolong its UFH-neutralizing effect in rats. This sustained anti-heparin activity enables R15 to avoid the heparin rebound associated with PS. Meanwhile, R15 appears safer than PS when present in excess *in vivo*. *In vitro*, although supratherapeutic amounts of both PS and R15 prolonged the activated partial thromboplastin time (APTT), R15 is degraded within 2 h and the APTT normalizes, whereas excess PS maintained a persistently prolonged APTT [20]. Furthermore, R15 lacks the capacity of PS to induce explosive complement activation when UFH is in excess. On the basis of the current data, R15 appears to be a safer potential heparin antagonist than PS.

## Conclusion

We established an *in vivo* analytical method based on measuring anti-heparin activity for PS and R15 and conducted the corresponding methodological validation. This assay enables monitoring of the anti-heparin capacity of heparin

antagonists in rats. In addition, we performed pharmacokinetic studies of single-dose PS and three dose levels of R15 in rats. Our results show that, compared with PS, R15 markedly prolongs the duration of action in rats, which could prevent heparin rebound.

## Supporting information

**S1 File.** S1 Table. Standard curve of PS in blank plasma. S2 Table. Standard curve of R15 in blank plasma. S3 Table. The stability of PS plasma sample placed in room temperature (25°C) for 30 min (n = 6). S4 Table. The stability of PS plasma sample freeze-thaw three cycles in −20°C (n = 6). S5 Table. The stability of stock solution of PS for 1 week (n = 6). S6 Table. The stability of R15 plasma sample placed in room temperature (25°C) for 30 min (n = 6). S7 Table. The stability of R15 plasma sample freeze-thaw three cycles in −20°C (n = 6). S8 Table. The stability of stock solution of R15 for 1 week (n = 6). S9 Table. Dilution effects of varying concentrations of plasma samples of PS diluted 2-fold, 5-fold, 10-fold, 20-fold (n = 5). S10 Table. Dilution effects of varying concentrations of plasma samples of R15 diluted 2-fold or 100-fold (n = 5). S11 Table. Pharmacokinetic parameters of intravenous infusion administration with PS (300 U/kg) to individual Wistar rats (n = 6). S11 Table. Pharmacokinetic parameters of intravenous infusion administration with PS (300 U/kg) to individual Wistar rats (n = 6). S12 Table. The plasma concentration of PS after intravenous infusion administration with PS (300 U/kg) to individual Wistar rats. ND: Not determined. S13 Table. Pharmacokinetic parameters of intravenous infusion administration with R15 (2700 U/kg) to individual Wistar rats (n = 8). S14 Table. Pharmacokinetic parameters of intravenous infusion administration with R15 (900 U/kg) to individual Wistar rats (n = 8). S15 Table. Pharmacokinetic parameters of intravenous infusion administration with R15 (300 U/kg) to individual Wistar rats (n = 8). S16 Table. The plasma concentration of R15 after intravenous infusion administration with R15 (300 U/kg) to individual Wistar rats. ND: Not determined. S17 Table. The plasma concentration of R15 after intravenous infusion administration with R15 (900 U/kg) to individual Wistar rats. ND: Not determined. S18 Table. The plasma concentration of R15 after intravenous infusion administration with R15 (2700 U/kg) to individual Wistar rats. ND: Not determined.
(ZIP)

## Acknowledgments

We would like to thank NES (https://nesediting.com) for their assistance with English language editing.

## Author contributions

**Conceptualization:** Guifang Dou, Huimin Li, Tong Li.

**Investigation:** Zhiyun Meng.

**Methodology:** Ruolan Gu.

**Writing – original draft:** Huimin Li, Tong Li.

**Writing – review & editing:** Tong Li, Xiaoxia Zhu, Ruolan Gu, Hui Gan.

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
