## [Decision Letter · Decision Letter 0]

27 Jul 2025

Dear Dr. Dou,

Thank you for submitting your manuscript to PLOS ONE. After careful consideration, we feel that it has merit but does not fully meet PLOS ONE’s publication criteria as it currently stands. Therefore, we invite you to submit a revised version of the manuscript that addresses the points raised during the review process.

We look forward to receiving your revised manuscript.

Kind regards,

Vikash Kumar Kumar Dubey

Academic Editor

PLOS ONE

Journal Requirements:

3. To comply with PLOS One submissions requirements, in your Methods section, please provide additional information regarding the experiments involving animals and ensure you have included details on (1) methods of sacrifice, (2) methods of anesthesia and/or analgesia, and (3) efforts to alleviate suffering.

5. Please amend your list of authors on the manuscript to ensure that each author is linked to an affiliation. Authors’ affiliations should reflect the institution where the work was done (if authors moved subsequently, you can also list the new affiliation stating “current affiliation:….” as necessary).

Reviewers' comments:

Reviewer's Responses to Questions

**Comments to the Author**

1. Is the manuscript technically sound, and do the data support the conclusions?

Reviewer #1: Yes

Reviewer #2: Yes

2. Has the statistical analysis been performed appropriately and rigorously?

Reviewer #1: No

Reviewer #2: Yes

3. Have the authors made all data underlying the findings in their manuscript fully available?

Reviewer #1: Yes

Reviewer #2: Yes

4. Is the manuscript presented in an intelligible fashion and written in standard English?

Reviewer #1: Yes

Reviewer #2: Yes

Reviewer #1: In this interesting manuscript, the authors, Li, Li et al., describe the pharmacokinetics of R15 and compare it to PS. Although striking the manuscript requires some clarification and additional analyses before publication. My concerns are as follows:

1. In the introduction section describing R15, there are no citations. It makes it sound like R15 is being described in this manuscript for the first time. This must be rectified.

2. The authors say that the degradation products of R15 may also exhibit UFH antagonistic properties. There is no citation for this and therefore, it is unclear how that may be mechanistically possible.

3. Although the rats were maintained under LD cycles, the authors do not mention at what time of day the tail vein injections were given. Were these times the same for PS and R15? If not, these results are not comparable in any way. If they were given at the same time, what time was it? There is plenty of research showing that responses to pharmacologically active agents are dependent on the circadian time of the organism. Therefore, are the differences observed between PS and R15 a simple consequence of the internal circadian clock? Would their pharmacokinetics be the same under different times of injections?

4. While the tables show male and female rat data separately, the average dynamics in the plots are pooled over both sexes. I suggest averaging the male and female rats separately and showing them in the figures.

5. It appears that there is a much larger sex difference in PS responses than R15 responses. The authors are encouraged to discuss the implications of this.

6. The authors compute various parameters in the pharmacokinetics experiments, but do not carry out any statistical comparisons to show significant differences in PS vs R15 activity. These tests must be carried out and their results reported.

7. Typo: Line 507 says "Founding" instead of "Funding."

Reviewer #2: This study establishes an innovative bioanalytical method for quantitatively determining the total anti-heparin activity of protamine sulfate and synthetic peptide R15 in rat plasma, accompanied by a systematic pharmacokinetic investigation. The research design is rigorous, data presentation is comprehensive, and methodological validation is thorough. The findings hold significant value for developing novel heparin antagonists. While the overall quality of the paper is high, certain details require refinement:

1. The methodological validation process lacks assessment of matrix effects and recovery rates.

2. It is not necessary to present AUC values for individual rats, displaying the mean values of the 8 rats is sufficient.

3. Reference formatting must strictly adhere to the journal's Guide for Authors.

4. The following sections should be relocated to the Materials and Methods section: (1) Regarding drug dosage and concentration units; (2) Conversion formulas from U·mL⁻¹ to g·mL⁻¹

5. The expression "v/v" in line 190 should be formatted in italics.

**Do you want your identity to be public for this peer review?** For information about this choice, including consent withdrawal, please see our Privacy Policy

Reviewer #1: No

Reviewer #2: No

---

## [Author Response · Author response to Decision Letter 1]

30 Aug 2025

To Editor:

1. I ensured that my manuscript meets PLOS ONE's style requirements, including those for file naming.

2. I ensured that all the authors have an ORCID iD and that it is validated in Editorial Manager.

3. In my Methods section, I have provided additional information regarding the experiments involving animals and have included details on (1) methods of sacrifice, (2) methods of anesthesia and/or analgesia, and (3) efforts to alleviate suffering.

4. I have added the supporting information for Data Availability

5. I have amended the list of authors on the manuscript to ensure that each author is linked to an affiliation.

To reviewers:

1. In the introduction section describing R15, there are no citations. It makes it sound like R15 is being described in this manuscript for the first time. This must be rectified.

Response: Thank you for the correction. We added the appropriate citations for R15 in the Introduction. The insertion appears at line 84 in unmarked version, citing references 19 and 20.

2. The authors say that the degradation products of R15 may also exhibit UFH antagonistic properties. There is no citation for this and therefore, it is unclear how that may be mechanistically possible.

Response: Thank you for your question. In our preliminary experiments, we synthesized a series of poly-arginine (R3-R20), among which R9-R20 demonstrated UFH-neutralizing activity [1]. We also observed in vivo that R15 is metabolized by metallocarboxypeptidases into R12, R13, and R14 [2, 3], indicating that several metabolites of R15 retain anti-UFH activity. Because the article and thesis are in Chinese, we added the URL of article/ dissertation (in English) at the end of this response.

Those two data can be accessed at the following URL:

https://www.cnki.net/KCMS/detail/detail.aspx?dbcode=CMFD&dbname=CMFD202001&filename=1020017862.nh&uniplatform=OVERSEA&v=MHHoH8tovBdtzl0N6IB7YbVeuLQLemySAI8eDK1YEKEsUfEUMN4Vp4BtA-bKmQR8

https://www.cnki.net/KCMS/detail/detail.aspx?dbcode=CJFD&dbname=CJFDLAST2018&filename=GWYZ201804016&uniplatform=OVERSEA&v=bFArNJ0r4TvqGMw9zAj6-eRQo8YU1l-2mg6tqG1hikzscteGFPKEJ_h0lhSK9lH_

3. Although the rats were maintained under LD cycles, the authors do not mention at what time of day the tail vein injections were given. Were these times the same for PS and R15? If not, these results are not comparable in any way. If they were given at the same time, what time was it? There is plenty of research showing that responses to pharmacologically active agents are dependent on the circadian time of the organism. Therefore, are the differences observed between PS and R15 a simple consequence of the internal circadian clock? Would their pharmacokinetics be the same under different times of injections?

Response: Thank you for your valuable question. Pharmacokinetic studies of PS and R15 were initiated in parallel on the same day and at the same clock time. Before study initiation, animals underwent carotid artery cannulation under anesthesia and were monitored daily. Animals in poor condition were excluded prior to randomization. The PK experiments in vivo in rats of PS and R15 at 300 U�kg-1 were performed at approximately 9:00 AM. On the following day, at the same time, PK experiments were conducted for subsequent dose of R15.

4. While the tables show male and female rat data separately, the average dynamics in the plots are pooled over both sexes. I suggest averaging the male and female rats separately and showing them in the figures.

Responds: Thank you for the valuable suggestion. We redraw the figures and present sex-stratified plots for female and male, as well as pooled figures (Figs. 3-6).

5. It appears that there is a much larger sex difference in PS responses than R15 responses. The authors are encouraged to discuss the implications of this.

Response Thank you for the great suggestion. The following discussion has also been incorporated into the manuscript at line 390 to 405 in unmarked version.

The observed male-female disparity in PS pharmacokinetics in rats is most plausibly attributable to differences in systemic clearance capacity. First, in humans, PS exposure is higher in men and systemic clearance is lower than in women [4, 5], suggesting that the sex difference seen in rats likely reflects an intrinsic, sex-dependent biological mechanism rather than inter-individual variation among rats. In rats, PS is cleared predominantly by the kidneys [4]. At the cellular level, PS is filtered at the glomerulus and taken up by renal proximal tubular epithelium via receptor-mediated endocytosis-principally through the megalin/cubilin complex-followed by lysosomal processing [6, 7]. Renal tubular function also exhibits sex dimorphism: compared with males, female rats have a lower fractional reabsorption of sodium and water in the proximal tubule, resulting in a greater fraction of the filtered load being delivered to downstream segments [8]. In addition, a recent rat study demonstrated higher megalin protein abundance in the renal cortex of females than males, implying greater proximal tubular endocytic/processing throughput in females; consequently, positively charged small polypeptides such as PS are removed from plasma more rapidly, and because the kidney effectively “captures” PS early, females exhibit smaller Vd and MRT [8, 9].

Additionally, we discovered an error in our previous calculation of the sex-stratified PK parameters for PS, which has now been corrected. We sincerely apologize for this oversight. These data are available in Dr. Li Tong’s 2021 doctoral dissertation (p. 126, Table 4.8). I will append the relevant data at the end of this response.

As the corrected data still demonstrate sex differences in PS pharmacokinetics in rats, we provided the above discussion accordingly.

Note: Due to an insufficient number of plasma concentration-time points in rats 3# and 14#, the corresponding pharmacokinetic parameters could not be accurately estimated; therefore, these animals were excluded from the calculation of mean pharmacokinetic values.

Dr. Li Tong’s 2021 doctoral dissertation can be accessed at the following URL:

https://www.cnki.net/KCMS/detail/detail.aspx?dbcode=CDFD&dbname=CDFDLAST2023&filename=1022636220.nh&uniplatform=OVERSEA&v=I-GbiIuliE5iOKh_Stq36ll8OG0QAkHeMBOUTYoj3-HorH6pJAHhKMxFOxb-ymk7

6. The authors compute various parameters in the pharmacokinetics experiments, but do not carry out any statistical comparisons to show significant differences in PS vs R15 activity. These tests must be carried out and their results reported.

Response: We thank the reviewer for the question and suggestions. We performed statistical comparisons of the pharmacokinetic parameters for PS and R15 at 300 U/kg; detailed results are provided at line 418 to 433 in unmarked version.

We compared the PK parameters of PS and R15 at the same dose of 300 U�kg-1. Across females, males, and pooled analyses, R15 showed statistically significant differences from PS in T₁/₂, AUC, CL, and MRT, except for male AUC and MRT and female CL. The most likely explanation is the small sample size and inter-individual variability leading to non-normal distributions. From the pooled data, compared with PS, R15 exhibits higher systemic exposure and residence (AUC and MRT), a longer T₁/₂, and a lower CL. The Vd values were not significantly different, indicating that the two compounds occupy similar initial distribution volumes; consequently, the observed differences in systemic exposure are driven primarily by variations in clearance rather than distribution. Consistent with the raw data, the T1/2 and MRT of R15 are approximately 13-fold and 19-fold those of PS, respectively; CL is about 1/20 that of PS; AUC is about 18-fold that of PS; and both Vd and Cmax are slightly lower for R15. These differences are mainly attributable to the fact that active metabolites of R15 retain pharmacological activity after degradation of the parent peptide, effectively mitigating heparin rebound caused by the rapid degradation of PS.

7. Typo: Line 507 says "Founding" instead of "Funding."

Response: Thank you for the correction. The typographical error in “Funding” has been corrected.

Reviewer #2: This study establishes an innovative bioanalytical method for quantitatively determining the total anti-heparin activity of protamine sulfate and synthetic peptide R15 in rat plasma, accompanied by a systematic pharmacokinetic investigation. The research design is rigorous, data presentation is comprehensive, and methodological validation is thorough. The findings hold significant value for developing novel heparin antagonists. While the overall quality of the paper is high, certain details require refinement:

1. The methodological validation process lacks assessment of matrix effects and recovery rates.

Response: Thank you for your suggestion. Our assay is a functional, chromogenic anti-FXa readout that quantifies the total anti-heparin activity of PS and R15 (parent plus active metabolites) after a defined protein-precipitation step and re-equilibration with a fixed UFH concentration. The method was fully validated for PS and R15 in accordance with Chinese bioanalytical guidance from Chinese Pharmacopoeia, confirming acceptable selectivity, precision and accuracy, stability, and dilution integrity. Therefore, LC-MS/MS–type “matrix effects” (ion suppression/enhancement) and absolute extraction recovery are not directly applicable to this ligand-binding/functional format. Instead, per common LBA validation practice, we assessed (i) selectivity across 10 individual rat plasma lots at LLOQ/ULOQ with blanks all “ND” and RE within acceptance, indicating no relevant matrix interference; (ii) dilution integrity/parallelism, showing commutability of endogenous matrix across 2–100-fold dilutions with RE and RSD <20%; and (iii) stability under room temperature hold, freeze–thaw, and stock-solution storage, all within pre-specified limits. These datasets collectively demonstrate that matrix constituents do not bias quantitation and that the sample-processing step preserves measurable activity.

2. It is not necessary to present AUC values for individual rats, displaying the mean values of the 8 rats is sufficient.

Response: We appreciate your suggestion. We have removed the plots of individual-rat AUC values and retained only the plots of the group mean across the eight rats (n=8).

3. Reference formatting must strictly adhere to the journal's Guide for Authors.

Response: We appreciate your valuable suggestion. We revised the reference formatting in accordance with the journal’s guidelines (Guide for Authors).

4. The following sections should be relocated to the Materials and Methods section: (1) Regarding drug dosage and concentration units; (2) Conversion formulas from U·mL⁻¹ to g·mL⁻¹

Response: We appreciate your valuable suggestion. We have relocated the following content to the Materials and methods section: (1) the description of drug dose and concentration units; and (2) the conversion formulas from U·mL⁻¹ to g·mL⁻¹.

5. The expression "v/v" in line 190 should be formatted in italics.

Response: We appreciate your suggestion. The notation “v/v” in manuscript has been formatted in italics.

References

1. Han Su DG, Meng Zhiyun, Zhu Xiaoxia, Gan Hui, Gu Ruolan, Wu Zhuona, Liu Taoyun, Li Jian, Zheng Ying, Li Tong, inventor; Institute of Transfusion Medicine, Academy of Military Medical Sciences, PLA of China /Han, Su (co-applicant), assignee. Protamine peptidomimetic, and pharmaceutically acceptable salts and use thereof. EP patent EP 2982680 A1. 2016 2014-04-01.

2. Yongle Z. The stability and metabolism of protamine peptide analog in vitro. China: Shandong First Medical University; 2018.

3. Zhu Yongle LT, Meng Zhi yun, Gan Hui, Zhu Xi aoxia, Dong Xiaona, Qi Yongxiu, Wang Decai, Dou Guifang The in vitro stability and proteolytic enzyme of synthesized protamine-analog peptide R15. J Int Pharm Res. 2018;45(4):295-300. doi:10.13220/j.cnki.jipr.2018.04.009.

4. Delucia III AW, Thomas W Kadell, Amy M Wrobleski, Shirley K VanDort, Marcian Stanley, James C. Tissue distribution, circulating half-life, and excretion of intravenously administered protamine sulfate. ASAIO journal. 1993;39(3):715-8.

5. Butterworth J, Lin YA, Prielipp R, Bennett J, James R. The pharmacokinetics and cardiovascular effects of a single intravenous dose of protamine in normal volunteers. Anesthesia & Analgesia. 2002;94(3):514-22. doi:10.1097/00000539-200203000-00008.

6. Nagai J, Komeda T, Katagiri Y, Yumoto R, Takano M. Characterization of protamine uptake by opossum kidney epithelial cells. Biological and Pharmaceutical Bulletin. 2013;36(12):1942-9. doi:10.1248/bpb.b13-00553.

7. Akour AA, Kennedy MJ, Gerk PM. The role of megalin in the transport of gentamicin across BeWo cells, an in vitro model of the human placenta. The AAPS journal. 2015;17(5):1193-9. doi:10.1208/s12248-015-9778-9.

8. McDonough AA, Harris AN, Xiong L, Layton AT. Sex differences in renal transporters: assessment and functional consequences. nature reviews nephrology. 2024;20(1):21-36. doi:10.1038/s41581-023-00757-2.

9. Tsuji S, Hasegawa-Izaki A, Ogawa B, Yamada H. Testosterone contributes sex differences of urinary biomarkers for nephrotoxicity in rats. The Journal of Toxicological Sciences. 2025;50(8):413-24. doi:10.2131/jts.50.413.

---

## [Editor Report · Decision Letter 1]

16 Sep 2025

Establishment of an in vivo analytical method for detecting total anti-UFH activity and pharmacokinetic study in PS and R15 in rats

PONE-D-25-33653R1

Dear Dr. Dou,

We’re pleased to inform you that your manuscript has been judged scientifically suitable for publication and will be formally accepted for publication once it meets all outstanding technical requirements.

Kind regards,

Vikash Kumar Kumar Dubey

Academic Editor

PLOS ONE
---

## [Editor Report · Acceptance letter]

PONE-D-25-33653R1

PLOS ONE

Dear Dr. Dou,

I'm pleased to inform you that your manuscript has been deemed suitable for publication in PLOS ONE. Congratulations! Your manuscript is now being handed over to our production team.

Kind regards,

on behalf of

Professor Vikash Kumar Kumar Dubey

Academic Editor

PLOS ONE